# Deformation and exhumation in thick continental crusts

# induced by valley incision of elevated plateaux

3

1

2

- Thomas Geffroy<sup>1</sup>, Philippe Yamato<sup>1,2</sup>, Philippe Steer<sup>1,2</sup>, Benjamin Guillaume<sup>1</sup>, Thibault 4
- 5 Duretz<sup>3</sup>
- 6 7 8 9 <sup>1</sup>Univ.Rennes, CNRS, Géosciences Rennes UMR 6118, 35000 Rennes, France
  - <sup>2</sup>Institut universitaire de France (IUF)
- <sup>3</sup>Institut für Geowissenschaften, Goethe-Universität Frankfurt, Frankfurt, Germany

10 Correspondence to: Philippe Yamato (philippe.yamato@univ-rennes.fr)

11

#### 12 **Short summary**

13 14

- While erosion's role in mountain building is well known, deformation from valley incision in inactive regions is
- less understood. Using our numerical models, we show that incision alone can cause significant crustal deformation
- and drive lower crust exhumation. This is favored in areas with thick crust, weak lower crust, and high plateaux.
- Our results show surface processes can reshape Earth's surface over time.

#### **Abstract**

20 21

crustal exhumation.

Surface processes such as erosion and sedimentation play a critical role in crustal deformation, particularly in actively deforming orogenic belts. While these processes have been extensively studied in large-scale erosive and tectonically active regions, the specific effects of valley incision on crustal deformation, especially in tectonically inactive regions, remain poorly understood. In this study, we hypothesize that crustal deformation induced by valley incision is primarily governed by three parameters: incision velocity, crustal thickness, and the elevation difference between the plateau and the valley base level. Using two-dimensional thermo-mechanical models, we investigate the influence of valley incision on crustal deformation and exhumation by varying these parameters. Our results show that valley incision alone can induce significant crustal deformation, associated with lateral viscous flow in the lower crust leading to near-vertical channel flow and extensional brittle deformation in the upper crust below the valley. This deformation leads to lower crust exhumation, within a 10 Myr timeframe, if crustal thickness is greater than 50 km, the initial plateau elevation is greater or equal to 2 km, and the long-term effective erosion rate exceeds 0.5 mm.yr<sup>-1</sup>. Furthermore, while the onset of lower crust exhumation is primarily controlled by the initial plateau elevation, the total amount of exhumed lower crust after 10 Myr strongly increases with the initial thickness of the lower crust which favors viscous flow. Moreover, natural systems that exhibit the required crustal thickness, plateau elevation, and erosion rates for lower crustal exhumation, as highlighted in our models, also demonstrate active lower crustal exhumation, as it is the case in regions such as Nanga Parbat and Namche Barwa. These findings offer new insights into the coupling between surface processes and deep crustal dynamics, highlighting the potential for valley incision to drive substantial crustal deformation and promote lower

# 1 Introduction

40

41

77

78

tectonically driven exhumation difficult.

43 Surface processes represent a pivotal element in the evolution of mountain ranges (e.g., Beaumont et al., 1992; 44 Koons, 1990). While tectonic forces increase topographic relief and slope, numerous modelling studies have 45 shown that surface processes and the associated mass redistribution exert a significant influence on tectonic 46 processes (Avouac and Burov, 1996; Braun and Yamato, 2010; Steer et al., 2014; Sternai et al., 2019; Thieulot et 47 al., 2014; Vernant et al., 2013; Willett, 1999; Wolf et al., 2022). Previous studies have demonstrated the critical 48 role of erosion in controlling deformation within active orogens (e.g. Thieulot et al., 2014; Wolf et al., 2022), 49 triggering complex feedback mechanisms, such as isostatic rebound (e.g., Sternai et al., 2019), or the setting up 50 of large lower crustal flow (Beaumont et al., 2001), involving mechanical, thermal and time-dependent processes. 51 These mechanisms enhance strain localization through intensified erosion, which in turn promotes increased rock 52 uplift and accentuates orogen asymmetry (e.g. Braun and Yamato, 2010; Thieulot et al., 2014; Wolf et al., 2022). 53 However, these studies specifically focused on active orogens which limits a comprehensive understanding of the 54 isolated influence of erosion on crustal deformation, particularly in regions without active tectonics. Moreover, 55 these studies do not specifically focus on the impact of valley incision on crustal deformation. 56 Nevertheless some studies demonstrate that valley incision can significantly influence topography through the 57 response of lithosphere deformation to erosion (Montgomery & Stolar, 2006; Simpson, 2004; Zeitler et al., 2001). 58 Baiadori et al. (2024) show that, even in the absence of regional deformation, the incision of fluvial valleys can 59 lead to lithospheric bending and significant change in landscape dynamics. However, these studies have not 60 explored the role of valley incision on large-amplitude deformation, accounting for visco-elasto-plastic rheologies, 61 over timescales relevant for orogen building. Recently, Yang et al. (2023) proposed that the erosion of wide valleys 62 (50 km), could drive significant crustal deformation, leading to the development of substantial relief (1–2 km) and, 63 in extreme cases, the exhumation of lower crustal material. While this study presents important findings, the 64 inclusion of a horizontal shortening rate of approximately 1.6 mm.yr<sup>-1</sup> limits our ability to isolate the effects of 65 valley incision on crustal deformation. 66 If rivers influence crustal deformation, different river systems should exert distinct impacts depending on various 67 parameters, such as incision velocity, crustal thickness, or base level. Specifically, when considering a plateau in 68 a non-convergent setting, we expect the difference of altitude between the plateau and the base level (what we 69 refer to as "erosion potential" here in after) to be a main controlling factor of the induced tectonic response. Indeed, 70 we hypothesize that a river with a low erosion potential would have a limited impact on crustal deformation, 71 whereas a river on a high plateau, implying a high erosion potential, could induce larger crustal deformation 72 (Figure 1). We expect the deformation to be accommodated by a flow of the viscous lower crust, leading to the 73 formation of high relief and, in extreme cases, to the exhumation of lower crustal material at the surface (Figure 74 1). 75 The potential for valley incision to drive the exhumation of lower crustal material has been previously debated in 76 specific natural settings, such as the Nanga Parbat–Haramosh Massif and the Namche Barwa Massif (Zeitler et al.,

2001). However, the tectonic complexity of these regions makes the distinction between erosionally and

In this study, we therefore focus on a scenario in which regional tectonic forces are absent, ensuring that all observed crustal deformation results solely from valley incision. This does not imply that our models are limited to inactive tectonic settings (thick continental crust indeed generally requires important shortening). However, as our objective is to isolate the specific impact of valley incision on crustal deformation, we voluntary imposed a fixed horizontal velocity boundary conditions set to nearly 0 (i.e., 0.015 mm.yr<sup>-1</sup>). Applying convergence would modify our model behavior and would make difficult to quantify what is due solely to the effect of the river incision. To systematically investigate the impact of valley incision on crustal deformation, exhumation, and their feedback on valley morphology, we employ a two-dimensional (2D) thermo-mechanical numerical model incorporating a visco-elasto-plastic rheology and surface processes consisting in a simple erosion law coupling a source term for river incision with diffusion (see Section 2.1). Our objective is to gain a deeper understanding of the mechanical response of the lithosphere to valley incision and evaluate the potential for this process to drive the exhumation of lower crustal material, as well as whether such exhumation can occur within a geological timeframe of less than 10 Myr. This value of 10 Myr was chosen as representative for the upper limit of river persistence at a fixed location over geological timescale. Examples of rivers maintaining the same course for more than 10 Myr are indeed extremely rare in nature (see section 4.3). Moreover, natural examples of valley incision associated with lower crust exhumation indicate that such exhumation typically occurs within less than 10 Myr (e.g., Nanga Parbat and Namche Barwa; Zeitler et al., 2001). Therefore, we assume that if this process does not take place within a 10 Myr timeframe in our model, it is unlikely to be geologically realistic.

Figure 1: Schematic representation of the impact of a valley incision on crustal deformation, depending on the erosion potential (see text for details).

#### 2 Materials and Methods

# 2.1 Code description

9798

100

103104

106

Numerical simulations are performed using the 2D thermo-mechanical code MDoodz (Duretz et al., 2021; Yamato et al., 2015). This numerical code was applied to the study of grain- (e.g., Luisier et al., 2023; Yamato et al., 2019), to lithospheric-scale (e.g., Auzemery et al., 2022; Candioti et al., 2022; Poh et al., 2020; Porkoláb et al., 2021)

geological processes. It accounts for visco-elasto-plastic rheologies (Duretz et al., 2020) and a true free surface (Duretz et al., 2016). Based on the marker-in-cell method (Gerya & Yuen, 2003; Harlow & Welch, 1965), MDoodz employs the finite difference method, to solve the governing equations of momentum (1), mass conservation (2), and heat (3) on a staggered-grid. These equations are expressed as follows:

$$\frac{\partial \tau_{ij}}{\partial x_i} - \frac{\partial P}{\partial x_i} = -\rho^{\text{eff}} g_i, \tag{1}$$

$$\frac{\partial v_i}{\partial x_i} = 0, \tag{2}$$

$$\rho^{\rm eff} C_{\rm P}^{\rm eff} \frac{DT}{Dt} = \frac{\partial}{\partial x_i} \left( k \frac{\partial T}{\partial x_i} \right) + Q_{\rm r} + Q_{\rm SH}, \tag{3}$$

- Material properties are stored on particles that are advected through time using a fourth-order Runge-Kutta scheme.
- The evolution of the density field is described by the following equation:

$$\rho = \rho_0 (1 - \alpha (T - T_0)) (1 + \beta (P - P_0)), \tag{4}$$

where  $\rho_0$  is the reference density,  $\alpha$  is the thermal expansivity,  $\beta$  is compressibility,  $T_0$  and  $P_0$  are the reference temperature and pressure, set to 0°C and 10<sup>5</sup> Pa, respectively. The initial temperature field is obtained by solving the equilibrium heat equation using the reference thermal parameters (Table 1), except for the lithospheric mantle, for which the conductivity has been artificially set to a very high value to obtain an adiabatic asthenosphere. The Courant condition is set at 0.25 for all the simulations.

- To prevent numerical instabilities, we impose a minimum viscosity threshold of  $10^{18}$  Pa.s in the model and use a free surface stabilization algorithm (Duretz et al., 2011). This threshold is set to avoid viscosity variations exceeding seven orders of magnitude within the model. It should be noted that  $10^{18}$  Pa.s is still higher than the viscosity of magmas ( $\sim 10^2 10^{14}$  Pa.s; (e.g., Dingwell, 2006)) or partially molten rocks ( $\sim 10^{17} 10^{18}$  Pa.s; (e.g., Molitor et al., 2024)).
- To account for a visco-elasto-plastic rheology, the deviatoric strain rate tensor is additively decomposed such as:

$$\dot{\varepsilon}_{ij} = \dot{\varepsilon}_{ij}^{\ v} + \dot{\varepsilon}_{ij}^{\ e} + \dot{\varepsilon}_{ij}^{\ p}, \tag{5}$$

where the exponents v, e, and p correspond to the viscous, elastic, and plastic (frictional) components. We detail in Appendix A each component of the strain rate. Partial melting is also included in the numerical code (see Appendix B).

At a specific location, the vertical erosion of a plateau can occur through river or glacier incision, modulated by climatic conditions, upstream conditions (e.g., water and sediment fluxes) and local slope. In addition, the lateral erosion of the valley wall can occur through multiple processes, including river lateral mobility and hillslope destabilization, mass wasting processes, peri-glacial processes, soil production and creep (e.g., Brocard and van der Beek, 2006; Hancock and Anderson, 2002; Martin et al., 2011). Accounting for all these different and interrelated processes in a single numerical model represents a challenge, clearly out of the scope of this manuscript. Instead, we rely on the use of a simple erosion law that combines a constant incision velocity imposed at the top of the lithosphere, to simulate vertical incision by glaciers or rivers, with a topographic diffusion coefficient to simulate lateral valley erosion and sedimentation. In our models, the incision rate ( $V_i$ ) remains constant as long as the valley floor remains above the river base level, set at 0 km elevation. Once this base level is reached,  $V_i$  is set to 0 and remains so unless the elevation increases above this base level. In the following, we impose a prescribed valley incision velocity  $V_i$  at the model center over an initial width  $W_i$ . In addition, diffusion is applied to surface, considering a constant diffusivity coefficient (K). The equation controlling the time-evolution of elevation h of the free surface is therefore:

$$\frac{dh}{dt} = \frac{\partial}{\partial x} \left( K \frac{\partial h}{\partial x} \right) + V_i. \tag{6}$$

#### 2.2 Reference model design

To define the reference model set-up, we first compiled global data on crustal thickness (Crust 1.0 model; Laske et al., 2013) and Earth continental surface elevation (ETOPO1; NOAA, 2009). Based on these datasets, we constructed a diagram illustrating the correlation between both parameters (Figure 2a). The reference model was then selected to represent an intermediate position in the observed distribution.

The reference model includes a crust with a thickness (H) of 55 km and an initial plateau elevation ( $h_P$ ) of 3 km (Figure 2a). The model domain used in this study is 500 km wide and 160 km deep. The reference model therefore includes a continental lithosphere made of a 55 km thick continental crust resting on top of a 65 km thick lithospheric mantle and 30 km of asthenospheric mantle (Figure 2b). We perform simulations at 1 km resolution for the full parametric study, as resolution tests show good convergence for resolutions lower than 1 km (see Figure S1).

The incision width  $W_i$  in our study has been set at a constant value of 1 km. In the reference model, we apply an incision velocity ( $V_i$ ) of 30 mm.yr<sup>-1</sup> and a topographic diffusion coefficient (K) of 3.0 × 10<sup>-6</sup> m<sup>2</sup>.s<sup>-1</sup>. The value of K lie in the range of previous numerical modeling studies involving surface processes (e.g., Avouac & Burov, 1996; Munch et al., 2022; Yamato et al., 2008). To avoid material loss, we compensate the volume loss due to the incision of the free surface by the addition of asthenospheric mantle material evenly distributed across the model base. The bottom, right, and left boundaries are free-slip boundaries.

The temperature at the top and lithosphere-asthenosphere boundaries are maintained at 0 and 1330°C, respectively. The asthenosphere is assumed to be adiabatic. Heat flux is set to zero at the right and left boundaries of the model (Neumann boundary conditions). The initial brittle/ductile transition occurs at 6 km of depth for the reference model (Figure 2b). We define the boundary between the upper and lower crust based on the brittle/ductile transition

of the Westerly Granite rheology. In our models, the lower crust corresponds to the portion of the crust located beneath this transition zone (represented in green in Figure 2b) and is considered as the ductile portion of the crust. The parametrization of the reference model leads to conditions favorable to crustal convection, with a Rayleigh number of ~ 1×10<sup>5</sup>.

All material parameters are presented in Table 1 for the reference model.

Figure 2: a) Continental crustal thickness vs. elevation from natural data. Data for crustal thickness are from the Crust 1.0 model (Laske et al., 2013). Data for global earth altitude are from ETOPO1 (NOAA, 2009). Black points represent the combination of plateau elevation and crustal thickness used in the parametric study, and black star represent the reference model. The black line represents the linear relationship between crustal thickness and elevation assuming that this relationship is only driven by isostasy and using the material properties of our models (see Table 1). b) Initial model setup for the reference model. The horizontal limit between the red and green layers in the continental crust corresponds to the depth of the initial peak of strength in the crust that we define as the limit between the upper and

the lower crust (considering a homogeneous strain rate of  $10^{-18}$  s<sup>-1</sup>). White lines represent each  $200^{\circ}$ C isotherm. Black arrows indicate the input of material to balance the amount of material removed by the valley incision (see text for details). Initial topography (Plateau elevation) is here set to 3 km and the valley base level is at 0 km. c) Yield strength profile from the surface to 70-km depth, considering a homogeneous strain rate of  $10^{-18}$  s<sup>-1</sup>. d) Temperature profile from the surface to 150-km depth.

|                                                       | Crust                      | Lithospheric and astenospheric mantle |                                |                        |
|-------------------------------------------------------|----------------------------|---------------------------------------|--------------------------------|------------------------|
| Rheological and materials<br>parameters               |                            |                                       |                                |                        |
| Material                                              | Westerly Granite           | Dry Olivine dislocation creep         | Dry Olivine diffusion<br>creep | Olivine Peierls creep  |
| n                                                     | 3.3                        | 3.5                                   | 1                              | -                      |
| A (Pa <sup>-n</sup> .s <sup>-1</sup> )                | 3.1623 × 10 <sup>-26</sup> | 1.1 × 10 <sup>-16</sup>               | 1.5 × 10 <sup>-15</sup>        | -                      |
| Q (J.mol <sup>-1</sup> )                              | 1.865 × 10 <sup>5</sup>    | 5.3 × 10 <sup>5</sup>                 | 3.75 × 10 <sup>5</sup>         | -                      |
| m                                                     | -                          | -                                     | 3                              | -                      |
| d (m)                                                 | -                          | -                                     | 2 × 10 <sup>-3</sup>           | -                      |
| G (Pa)                                                | 3.0 × 10 <sup>10</sup>     | 3.0 × 10 <sup>10</sup>                | 3.0 × 10 <sup>10</sup>         | -                      |
| C (MPa)                                               | 50                         | 50                                    | 50                             | -                      |
| φ (°)                                                 | 30                         | 30                                    | 30                             | -                      |
| Crustal Thickness (km)                                | 60                         | -                                     | -                              | -                      |
| k (W.m <sup>-1</sup> .K <sup>-1</sup> )               | 2                          | 3.2                                   | 3.2                            | -                      |
| $\rho_o$ (kg.m <sup>-3</sup> )                        | 2800                       | 3260                                  | 3260                           | -                      |
| С <sub>Р</sub> (J.kg <sup>-1</sup> .K <sup>-1</sup> ) | 1050                       | 1050                                  | 1050                           | -                      |
| Q <sub>r</sub> (W.m <sup>-3</sup> )                   | 1.0 × 10 <sup>-6</sup>     | 1.0 × 10 <sup>-10</sup>               | 1.0 × 10 <sup>-10</sup>        | -                      |
| α (K <sup>-1</sup> )                                  | 3.0 × 10 <sup>-6</sup>     | 3.0 × 10 <sup>-6</sup>                | 3.0 × 10 <sup>-6</sup>         | -                      |
| β (Pa <sup>-1</sup> )                                 | 1.0 × 10 <sup>-11</sup>    | 1.0 × 10 <sup>-11</sup>               | 1.0 × 10 <sup>-11</sup>        | -                      |
| η <sub>νp</sub> (Pa.s)                                | 1.0 × 10 <sup>21</sup>     | 1.0 × 10 <sup>21</sup>                | 1.0 × 10 <sup>21</sup>         | -                      |
| Q Peierls (kJ.mol <sup>-1</sup> )                     | -                          | -                                     | -                              | 540                    |
| σ <sup>Peleris</sup> (Pa)                             | -                          |                                       | -                              | 8.5 × 10 <sup>9</sup>  |
| E <sup>Peleris</sup> (s <sup>-1</sup> )               |                            |                                       |                                | 5.7 × 10 <sup>11</sup> |
| 9                                                     | -                          | -                                     |                                | 2                      |
| γ                                                     | -                          | -                                     | -                              | 0.1                    |
| S                                                     |                            |                                       |                                | 1                      |

Table 1: Rheological and material parameters values used in our study for the reference model. The dislocation creep parameters used for the crust are those for Westerly Granite (Hansen & Carter, 1983). For the lithospheric and asthenospheric mantle, the chosen dislocation and diffusion creep parameters are those for Dry Olivine (Hirth & Kohlstedt, 2003) and the chosen Peierls creep are those for Olivine (Evans & Goetze, 1979; Kameyama et al., 1999). For elasticity, the shear modulus G is 30 GPa for all materials, which is in the range of values proposed for natural rocks at lithospheric depths (e.g., Turcotte & Schubert, 2002). Friction angle ( $\varphi$ ) and cohesion (C) are set to be representative of Byerlee's law (J. Byerlee, 1978). Values of  $Q_r$  are from Jaupart & Mareschal, (2021) and Rudnick & Gao, (2003).

#### 2.3 Parametric analysis

Two different systematic studies were performed from this reference model.

(1) We systematically vary (i) crustal thickness (*H*) from 40 to 65 km (5 km increments) and (ii) initial plateau elevation (*h*<sub>P</sub>) from 1 to 5 km (1 km increments), following their observed natural correlation (Figure 2a). We assume that, by using constant crustal and mantle densities and excluding horizontal velocities at the lateral boundaries of our models, isostasy imposes a linear correlation between crustal thickness and plateau elevation. This relationship implies that a single plateau elevation corresponds to each crustal

thickness (Figure 2a). However, to explore a broader range of geodynamic scenarios and to capture the diversity of natural systems, we deliberately extend our models beyond this constraint. This approach allows us to simulate configurations where thick crust is associated with anomalously low/high plateau elevations, cases that cannot be explained by isostasy alone. Relying strictly on the isostatic relationship would prevent us from independently assessing the roles of crustal thickness and plateau elevation. Nonetheless, most of our models, particularly the reference model, remain close to this linear relationship (Figure 2a).

We use the same incision velocity (Vi) of 30 mm.yr<sup>-1</sup> used for the reference model.

(2) We perform models with two different incision velocities (*V*<sub>i</sub>) of 10 and 50 mm·yr<sup>-1</sup> considering the same combination of crustal thickness and initial elevation as for the previous systematic study.

A total of 48 models were performed. All models with their respective parameters are available in Table S1.

#### 2.4 Model metrics

For each model, we design a set of metrics to quantify their evolution in terms of mechanical strength, temperature, exhumation and topography. To characterize valley morphology, we compute the valley maximum relief ( $\Delta h$ ), corresponding to the difference between the maximum ( $h_{\text{max}}$ ) and minimum ( $h_{\text{min}}$ ) elevation of the model surface (Figure 3a and 3b). We also compute the effective erosion rate ( $E_{\text{eff}}$ ) in the center of the valley by calculating the time required to remove successive2-km thick layers of material eroded by valley incision at the center of the valley. From this, we deduce the cumulative amount of vertical exhumation at different time step in the center of the valley. Because of lateral topographic diffusion, the incision velocity  $V_i$  does not directly correspond to the effective erosion velocity in the center of the valley as  $E_{\text{eff}}$  depends on both  $V_i$  and K. Hence, in the reference model for instance (with  $V_i = 30 \text{ mm.yr}^{-1}$ ), the mean  $E_{\text{eff}}$  value in the center of the valley is  $0.8 \pm 0.25 \text{ mm.yr}^{-1}$  between 1 and 10 Myr of model simulations (Figure 3c).

# 3 Results

#### 3.1 Reference model

# 3.1.1 Valley morphology and effective erosion

- The valley relief ( $\Delta h$ ) evolves over time, reaching a maximum value of 3.5 km after 4.5 Myr (Figure 3b). After this peak,  $\Delta h$  decreases almost linearly until the end of the simulation, reaching 3.2 km after 10 Myr. Notably, half of the total relief increase occurs within the first 1 Myr, emphasizing the transient nature of the topographic evolution under a prescribed incision rate. The primary driver of relief development is the decrease in  $h_{\min}$ , which decreases by 2.8 km after 4.5 Myr, rather than an increase in  $h_{\max}$ , which rises by only 700 m over the initial plateau elevation during the same period (Figure 3b).
- After a sharp decrease during the first 1 Myr,  $h_{min}$  decreases gradually, particularly between 1.5 and 3 Myr.
- Between 3 and 5 Myr,  $h_{\min}$  stabilizes at approximately 200 m before increasing again, reaching 600 m by the end
- of the simulation (10 Myr). In contrast,  $h_{\text{max}}$  increases almost linearly over the first 3 Myr, reaching 3.6 km, before
- gradually rising to a stable value of 3.8 km after 6 Myr (Figure 3b).

A bump of 100 to 200 m appears in both  $h_{\min}$  and  $h_{\max}$  curves after ~1.5 Myr, corresponding to the initiation of crustal convection (Fig. 4), which temporarily increases both values (Figure 3b).

The effective erosion rate ( $E_{\rm eff}$ ) follows a different evolution. It starts at  $\sim 9.5$  mm.yr<sup>-1</sup> during the first 100 kyr, rapidly decreases to reach  $\sim 1$  mm.yr<sup>-1</sup> after 1 Myr, and remains nearly constant at  $\sim 0.8$  mm.yr<sup>-1</sup> afterwards (Figure 3c).

Figure 3: (a) Topographic profile of the reference model after 1 Myr of simulation. Measurement of the relief ( $\Delta h$ ) and of the maximum ( $h_{\rm max}$ ) and minimum ( $h_{\rm min}$ ) altitude are indicated on the profile. (b) Time evolution of  $\Delta h$ ,  $h_{\rm max}$  and  $h_{\rm min}$  for the reference model. The black triangle represents the moment when the lower crust is exhumed to the surface. The vertical dotted line represents when the initiation of the crustal convection starts. (c) Time evolution of the effective erosion rate at the center of the valley ( $E_{\rm eff}$ ) for the reference model.

# 3.1.2 Crustal deformation and exhumation

277

Due to the presence of crustal convection, which obscures a clear visualization of the deformation patterns, our analysis focuses exclusively on the upper crustal layers unaffected by convective motion. Valley incision induces crustal deformation and alters crustal strength. Over the 10 Myr of the simulation, the reference model exhibits significant upward motion of the lower crust, with 3.5 km of exhumation to the surface (Figure 4a). The lower crust reaches the surface after 4.6 Myr (Figure 3b). The upward motion of the lower crust is accompanied by the upward advection of isotherms, at a slower rate due to heat diffusion (Figure 4a and S2a). Throughout the simulation, the Moho depth remains largely stable, with only minor variations associated with the onset of crustal convection. Crustal motions result in a non-uniform spatial distribution of strain rates (Figure 4b, left). Over time, strain rates remain largely diffuse within the non-partially melted portion of the crust. However, zones of elevated strain rates  $(>10^{-16} \text{ s}^{-1})$  develop within the viscous part of the non-melted crust, particularly beneath the center of the valley. After 5 Myr, this zone of localized high strain rates (up to  $10^{-15}$  s<sup>-1</sup>) reaches a width of approximately 10 km (Figure 4b. This high-strain rate zone in the viscous part of the crust is associated with brittle deformation in the upper crust (i.e., above 6 km of depth), exhibiting similar spatial dimensions and strain rate values (Figure 4b). The analysis of the stress field and of the direction of the maximum principal stress ( $\sigma_1$ ) suggest that the brittle

deformation in the center of the valley corresponds to extensional deformation, whereas the brittle zones further away from the valley center experiences compressional deformation (Figure 4b, right). In the viscous part of the crust, this is the opposite, with compressional deformation below the valley and extensional deformation 20-30 km away from the valley center (Figure 4b, right). As the simulation progresses, the region of viscous compressional deformation beneath the valley center migrates upward, reducing the extent of extensional deformation near the surface (Figure 4b, right). Elsewhere in the crust, the deformation pattern remains stable, with compressional deformation at the surface and extensional deformation at greater depths.

Partial melting is also important in the crust, as seen with the convection cells, and can reach values up to 60% of rocks melted in this lower part of the crust (Figure S2b). This fraction tends to decrease to 40 % after 10 Myr of simulation.

Figure 4: Time evolution of the reference model after 1, 5 and 10 Myr. (a) material and thermal evolution. White lines represent each 200°C isotherm. Color legend is as for Figure 2b. (b) left: Evolution of the second invariant of the strain rate. right: Evolution of the angle between the maximum principal stress ( $\sigma_1$ ) and the vertical axis).

## 3.2 Parametric study

## 3.2.1 Individual impact of crustal thickness and plateau elevation

Although crustal thickness (H) and plateau elevation (hP) are closely linked, a given crustal thickness can correspond to different plateau elevations, and conversely (Figure 2a).

Varying crustal thickness while maintaining a constant plateau elevation of 3 km has only a minor impact on the evolution of relief ( $\Delta h$ ) (Figure 5a). The main difference lies in the maximum relief, which reaches 3900 m for the thinner crust model (H = 50 km) and 3300 m for the thicker crust model (H = 60 km). This difference of 600 m is partly due to differences in the evolution of  $h_{min}$ . Indeed, in the thinner crust model, the river reaches its base level after 5.5 Myr, whereas in the thicker crust model,  $h_{min}$  only decreases down to 400 m after 3.5 Myr (Figure 5b).

Compared to the reference model, a model involving a thicker crust leads to a lower value of  $\Delta h$ . In such a model, the onset of exhumation to the surface of the lower crust is slightly earlier (4.6 Myr) and the river does not reach its base level (Figure 5a and 5b). In contrast, the presence of a thinner crust has the opposite effect, leading to higher  $\Delta h$  values, a delayed onset of lower crust exhumation to the surface (5.7 Myr), and a river reaching its base level (Figure 5a and 5b).

Instead, varying the initial plateau elevation, and hence the erosion potential, while maintaining a constant crustal thickness of 55 km results in larger differences. First, plateau elevation plays a key role in valley morphology. Models with a lower  $h_P$  value enable the river to reach the base level (Figure 5c and 5d). When  $h_P = 1$  km, the river reaches its base level in 200 kyr and remains at this altitude throughout the entire simulation (Figure 5d). This has a major impact on the relief, with  $\Delta h$  not exceeding 1.2 km and remaining at this value for the duration of the simulation (Figure 5c). A similar pattern is observed for the model where  $h_P = 2$  km, with differences in timing—the river reaches its base level at 1.1 Myr—and in the maximum  $\Delta h$  value, reaching 2.4 km. For  $h_P \geq 3$  km (i.e., including the reference model), the models behave identically with the river unable to reach its base level (Figure 5c and 5d). Consequently, increasing the plateau elevation above 3 km, while keeping the same combination of incision velocity and crustal thickness (i.e., 30 mm.yr<sup>-1</sup> and 55 km), produces the same outcome (Figure 5c, 5d and S3b).

In addition, the initial plateau elevation also significantly impacts lower crust exhumation. In the model where  $h_P$  = 1 km, the exhumation of the lower crust to the surface does not occur within the 10 Myr of the simulation (Figure 5c). In the model where  $h_P$  = 2 km, lower crust exhumation is possible but occurs at 8.5 Myr. This is later than in the reference model ( $h_P$  = 3 km) or for models with a higher plateau elevation ( $h_P$  > 3 km), which all show the onset of lower crust exhumation at 4.6 Myr.

#### Effect of crustal thickness

#### Effect of plateau elevation

Figure 5: Time evolution of  $\Delta h$  (left), and  $h_{\min}$  (right) for models with different values of H (a and b) or different values of  $h_P$  (c and d). For all models, the incision velocity ( $V_i$ ) is fixed at 30 mm.yr<sup>-1</sup>. Downward triangles represent the time when the lower crust is exhumed to the surface if it happens. Vertical dotted lines represent the time of the convection initiation within the crust.

#### 3.2.2 Combined impact of crustal thickness and plateau elevation

Looking at the entire parametric study conducted for an incision velocity of 30 mm.yr<sup>-1</sup>, the timing of lower crust exhumation appears to be mostly influenced by the initial plateau elevation rather than the crustal thickness (Figure 6a). Indeed, for models with  $h_P \ge 4$  km, exhumation occurs at 4.6-4.7 Myr, while models with  $h_P 

Figure 6: Time for the exhumation to the surface of lower crust with a)  $V_i = 30$  mm.yr<sup>-1</sup> and b)  $V_i = 50$  mm.yr<sup>-1</sup>. Values presented inside the circles are in Myr. The models with  $V_i = 10$  mm.yr<sup>-1</sup> are not presented due to the absence of lower crust exhumation in all models in the 10 Myr timeframe.

#### 3.2.3 Individual impact of the imposed incision velocity

Two different incision velocities (10 mm.yr<sup>-1</sup> and 50 mm.yr<sup>-1</sup>) were tested on the same set of model combinations as in the previous section with 30 mm.yr<sup>-1</sup>. Results show that the observed strain rate pattern as well as the deformation regime of the model remains almost the same as in the reference model (Figure S6).

Incision velocity also plays a crucial role in the timing of lower crust exhumation to the surface. With an incision velocity of 10 mm.yr<sup>-1</sup>, no exhumation occurs within the 10 Myr timeframe. With an incision velocity of 50 mm.yr<sup>-1</sup>, the conditions leading to lower crust exhumation to the surface within the 10 Myr timeframe (Figure 6b) remain similar to those obtained for an incision velocity of 30 mm.yr<sup>-1</sup>, even if exhumation of the lower crust occurs systematically earlier.

Incision velocity also has a significant impact on valley morphology and lower crust exhumation (Figure 7a). As expected, incision velocity directly influences the river ability to reach its base level. Keeping the same combination of crustal thickness and plateau elevation than in the reference model, the river fails to reach its base level in both the low-incision and reference models, whereas it successfully reaches its base level in the high-incision model (Figure 7b). Consequently, the evolution of  $\Delta h$  is closely tied to the capacity for the river to reach its base level. In the high-incision model,  $\Delta h$  reaches a plateau after around 1 Myr, which roughly corresponds to the time when the valley floor reaches the base level (Figure 7b). Thereafter,  $\Delta h$  remains nearly stable over time, exhibiting only a slight increase accompanied by minor oscillations, which correspond to fluctuations in  $h_{min}$ . In contrast, for the other two models,  $\Delta h$  evolution is therefore not limited by the river base level and evolves according to the combination of imposed surface processes and crustal response (Figure 7a).

Incision velocity also plays a crucial role in the timing of lower crust exhumation to the surface. In the low-incision model, no exhumation occurs within the 10 Myr timeframe, whereas in the high-incision model, exhumation begins earlier (at 3.6 Myr) than in the reference model.

# Effect of incision velocity

Figure 7: Time evolution of  $\Delta h$  (a), and  $h_{\min}$  (b) for models with different initial incision velocity. For all models, the values of  $h_P$  and H are fixed at 3 and 55 km, respectively. Downward triangles represent the time when the lower crust is exhumed to the surface if it happens. Vertical dotted lines represent the time of the convection initiation within the crust.

#### 4 Discussion

### 4.1 Effective erosion and comparison to natural river erosion

401 The evolution of effective erosion can be divided into two distinct phases (Figure S7): (1) the initial phase (< 1 402 Myr), when most models exhibit a rapid decrease in  $E_{\text{eff}}$  (Figures S7 and 3c) and (2) a stabilization phase (> 1 403 Myr), when effective erosion values tend to a steady state (Figures S7 and 3c). 404 During the initial phase, the mean value of E<sub>eff</sub>, ranging from 2 to 10 mm.yr<sup>-1</sup>, is highly dependent on the imposed 405 incision velocity, between 10 and 50 mm.yr<sup>-1</sup>, for  $h_P = 3$  km and H = 55 km. This phase derives from two different 406 behaviors of our models: (i) For models that do not reach the base level within 1 Myr (see Figure S3), the observed 407 decrease in Eeff is mainly driven by the action of topographic diffusion. In addition, this timescale of 1 Myr roughly 408 corresponds to the diffusion timescale  $t_K = (W/2)^2/4K$ , ranging between 0.6 and 1.6 Myr for a valley width (W) 409 varying between 30 and 50 km, respectively. Importantly, the decrease in effective erosion due to the topographic 410 diffusion does not change the total volume of eroded material over time. (ii) For models in which the valley reaches 411 the valley base level in less than 1 Myr (see Figure S3), the decrease in E<sub>eff</sub> is primarily controlled by the base 412 level. Once the valley reaches the base level, incision can no longer proceed, leading to a significant reduction in 413 effective erosion. During this initial phase, the tectonic response to incision can be considered negligible with 414 respect to the decrease in effective erosion. 415 During the stabilization phase, the mean E<sub>eff</sub> values range from 0.1 to 2 mm.yr<sup>-1</sup> (Figure 8). This steady-state phase 416 results from the tectonic response to incision. The development of a vertical crustal flow beneath the valley brings 417 deeper crustal material to the surface, uplifting the river above the base level and enabling its incision to persist. 418 Therefore, while  $V_1$  remains a key factor,  $h_P$  and H also influence long-term effective erosion values (Figure 8). 419 Natural valley incision rates can vary significantly due to differences in environmental settings, such as climatic 420 conditions, drainage basin area, or lithology. Another factor contributing to these variations is the timescale over 421 which incision rates are calculated or observed, as shorter- or longer-term measurements can lead to different 422 estimates of incision rates. (e.g., Finnegan et al., 2014; Gallen et al., 2015, 2015; Mills, 2000). Recent studies have 423 explored the relationship between incision rate values and the timescales over which they are measured, using 424 datasets of river incision rates from various locations on Earth (Nativ and Turowski, 2020). This relationship 425 indicates that for timescales shorter than 1 Myr, incision rates range from 0.01 to 5 mm yr<sup>-1</sup>, with an average of 426 0.4 mm yr<sup>-1</sup>. For timescales exceeding 1 Myr, incision rates range from 0.01 to 1 mm yr<sup>-1</sup>, with an average of 0.08 427  $mm yr^{-1}$ . 428 In our models, during the first Myr, effective erosion rates tend to be slightly higher than those observed in natural 429 settings for this timescale. Notably, models with incision velocities of 50 mm yr<sup>-1</sup> yield mean effective erosion 430 rates of 10 mm yr<sup>-1</sup> (Figure S7). Beyond 1 Myr, our models rarely produce effective erosion rates exceeding a 431 threshold of 1.1 mm yr<sup>-1</sup> (Figure 8). Only a limited number of simulations with an incision velocity of 50 mm yr<sup>-1</sup> 432 exceed this threshold (Figure 8c). 433

As explained in Section 2.5, the incision velocity does not directly correspond to the calculated effective erosion.

Figure 8: Mean values of the effective erosion velocity ( $E_{\rm eff}$ ) between 1 and 10 Myr for each models with, a)  $V_i = 10$  mm.yr<sup>-1</sup>, b)  $V_i = 30$  mm.yr<sup>-1</sup>, and c)  $V_i = 50$  mm.yr<sup>-1</sup>. Values presented inside the circles are in mm.yr<sup>-1</sup>. Black contours represent models with no exhumation of the lower crust in the 10 Myr timeframe. Red contours represent models presenting lower crust exhumation in the 10 Myr timeframe.

#### 4.2 Crustal deformation and lower crust exhumation

434435

436

437

438

439

440

442

Using visco-elasto-plastic models, Vernant et al. (2013) investigated the role of isostatic rebound induced by erosion in low-convergent mountain ranges. In their study, they performed some models without imposing tectonic boundary conditions and demonstrated that, in this context, erosion can induce regional uplift and contribute to maintaining mountain elevation. More importantly, the deformation regime observed in their models is very close to that obtained in our simulations, with erosion leading to horizontal extension in the uppermost kilometers of the crust and to horizontal compression at greater depths (Figure 4). This superficial extensional deformation is associated with plastic deformation and faulting in both models. Furthermore, the spatial pattern of crustal motion in both studies is similar, with intense vertical flow concentrated in the center of the model where erosion is applied, decreasing progressively away from the erosion zone, and ultimately leading to downward (negative) vertical flow at greater distances (Figure 9b). This vertical crustal flow is associated with the isostatic rebound induced by erosion, as mantle flow patterns in their study also highlight a vertical component. Similarly, our models show that valley incision can generate substantial vertical crustal flow, reaching values of nearly 0.8 mm yr<sup>-1</sup> at the center of the valley in the reference model (Figure 9a). An interesting feature observed in our models is that surface uplift can be decoupled from the lithospheric response, and that crust-mantle decoupling explains why the Moho remains stable while the lower crust migrates toward zones of lower pressure. This result can potentially explain the low effective elastic thickness (EET) often inferred from the isostatic response of the lithosphere to surface processes (Champagnac et al., 2007). At the global scale, EET values typically range between 30 and 80 km, reflecting a coupled mantle-crust system (e.g., Tesauro et al., 2012). However, in orogenic settings, EET values used to model uplift induced by erosion are generally significantly lower, often around 10 km (Champagnac et al., 2007). These low EET values likely reflect the mechanical behavior of the crust alone, rather than that of the entire lithosphere. This discrepancy helps to explain why erosion-induced uplift in orogens is best matched by low EET values. In such settings, the lower crust is fully ductile, enabling it to accommodate surface unloading without requiring a whole-lithosphere isostatic adjustment.

This vertical motion of the crust promotes the rise of low-viscosity material toward the surface (Figure 9b). Despite some differences in the processes responsible for initiating crustal motion, these similarities suggest that whether erosion is applied to an already elevated mountain range (Vernant et al., 2013) or results from localized valley incision (as in our case), the resulting crustal deformation exhibits comparable patterns. In the former case, erosion contributes to maintaining mountain elevation, while in our models, it promotes valley relief development and lower crust exhumation. The amount of exhumed lower crust at the center of the valley (as we refer as  $Q_{LC}$  hereafter), is highly sensitive to the input parameters of our models (Figure 10). Indeed, exhumation of the lower crust, reaching 6 km of depth below the model surface first depends on vertical incision of the valley bottom. Lower crust exhumation is first favored by the incision velocity  $V_i$  and the effective erosion rate  $E_{\text{eff}}$ , but is bounded by plateau elevation  $h_P$ , which sets the erosion potential. For instance, models with a 5 km high plateau relative to base level can exhume 5 over 6 km of depth simply by erosion. To reach lower crust material, valley incision is therefore not sufficient and a vertical uplift of the valley bottom, above the base level, is required to continue erosion. This second process, as previously discussed, depends on the upward velocity of viscous flow below the valley, and in turn on the effective viscosity of the lower crust. All other things being equal, increasing crustal thickness H, and in turn the ductile lower crust thickness, decreases this viscosity and increases the velocity of the upward flow (Figure 9b). A more rapid upward flow increases effective erosion rates and shortens the time required to exhume lower crust material. To illustrate the importance of the ductile lower crust thickness, we performed two additional models. In the first one, we reduced the radiogenic heat production of the crust, resulting in an initially colder thermal structure (i.e.,

substantially delays lower crustal exhumation, or in some cases, inhibits it entirely.

Therefore, if  $h_P$  and  $V_i$  mostly control the onset of lower crust exhumation, in particular for the most elevated plateaux, the amount of lower crust exhumation after a few million years is tightly linked to viscous flow of the lower crust in response to incision, and in turn to H and the ductile lower crust thickness.

800 °C at the Moho). In the second one, we applied a different rheology for the lower crust, using the Maryland

Diabase flow law (Mackwell et al., 1998) (see Figures S8 and S9 for thermal and rheological parameters). Both

configurations result in a stronger/thicker upper crust and a thinner ductile lower crust (Figures S8a and S9a).

These models show that lower crustal exhumation is significantly reduced under these conditions, with exhumation

occurring only after approximately 50 Myr of simulation time, at best (Figures S8b and S9b). These results

emphasize the critical role of the ductile lower crustal thickness in enabling exhumation: a thinner ductile layer

Figure 9: (a) Vertical velocity at z=-5 km for different timesteps for the reference model with H=55 km (left) and the model with H=65 km (right). (b) Viscosity distribution for the reference model (left) and the model with H=65 km (right) after 10 Myr. White arrows represent the velocity field. We only show velocity field within the non-partially melted part of the crust to enhance visibility and clarity.

Figure 10: Quantity of exhumed lower crust in the center of the valley ( $Q_{LC}$ ) for the models with  $V_i = 30$ mm.yr<sup>-1</sup> after (a) 5 Myr and (b) 10 Myr of simulation. Values presented inside the circles are in km.

# 4.3 Application to natural settings: the conditions required for valley incision to exhume lower crust

Our models allow us to determine the conditions leading to the exhumation of lower crust material when a continental crust is submitted to valley incision only. In this configuration, exhumation of lower crust is only possible:

- (1) for high initial plateau elevation (i.e. for a value of  $h_P \ge 2$  km over a 10 Myr timeframe). Over a 5 Myr timeframe, this threshold increases to 3 km.
- (2) for high crustal thickness (i.e. if H > 50 km over 10 Myr). Over 5 Myr, the threshold increases to 55 km.
- (3) for an erosion rate above 0.5 mm.yr<sup>-1</sup> over 10 Myr, with a threshold increasing to 0.8 mm.yr<sup>-1</sup> over 5 Myr.
- Using these criterions, we can compare our findings to natural river systems on Earth associated or not with lower crust exhumation (Figure 11). We emphasize that no natural setting perfectly matches the boundary conditions and setup of our models, which in turn leads to a certain degree of interpretation when comparing models with data.
- Rivers such as the Ebro and Potomac do not exhibit lower crust exhumation, which aligns well with our models when considering crustal thickness, elevation, incision rates, and duration of incision (Figure 11).
  - The Tsangpo and Indus rivers also align with our model predictions, as lower crust exhumation is observed in Namche Barwa and Nanga Parbat (Figure 11). However, given the intense tectonic forcing in these regions, with important convergence, we acknowledge that lower crust exhumation may also result from additional processes beyond valley incision. Nonetheless, our models provide a well-founded hypothesis regarding lower crust exhumation in these areas. A dedicated modeling study including horizontal convergence would be necessary to confirm this hypothesis, but such an investigation lies beyond the scope of this study.

The Yellow River, despite having crustal thickness and elevation within the predicted range for exhumation (Figure 11a), does not exhibit lower crust exhumation. This discrepancy is likely due to the short duration of incision, which began only 1.25 Myr ago (X. Wang et al., 2022) (Figure 11b). This underscores the critical role of incision duration in determining whether a river can induce lower crust exhumation.

Finally, the Grand Canyon is an outlier. It shows lower crust exhumation despite not meeting the required crustal thickness and plateau elevation predicted by our models (Figure 11a). The Grand Canyon presents a complex case and its formation remains debated, with some studies suggesting that regional uplift driven by mantle dynamics — known as epeirogeny — may have contributed to its development (Karlstrom and Timmons, 2012). Epeirogenic uplift corresponds to the vertical rise of large crustal regions without significant tilting, folding, or thrusting. This additional uplift, which is not accounted for in our models, could have contributed to increasing the plateau elevation, thereby enhancing the erosion potential. Moreover, the presence of a Proterozoic basement beneath the sedimentary cover (Karlstrom and Timmons, 2012; Shufeldt et al., 2010) suggests that the eroded lower crustal material in the Grand Canyon was potentially already located near the surface prior to the onset of incision, which would have facilitated its exhumation. Indeed, such a regional uplift, and the presence of old lower crust material close to the surface, could have enabled the Grand Canyon to exhume lower crustal material without necessarily meeting the specific crustal thickness, plateau elevation, and incision rates required in our models.

Figure 11: Comparison of our models to natural river systems for a) elevation and crustal thickness and b) incision rates and time of incision. The dotted red and orange lines represent the limit between models exhuming or not exhuming lower crust in the 10 Myr and 5 Myr timeframe, respectively. Red points represent river systems with lower crust exhumation. Different river systems and references used are: Eb = Ebro river (Regard et al., 2021), Po = Potomac river (Reusser et al., 2004), GC = Grand Canyon (Darling et al., 2012; Karlstrom et al., 2021; Pederson et al., 2002), Ts = Tsangpo river (near Namcha Barwa) (Finnegan et al., 2014; Koons et al., 2022; Wang et al., 2017), Ye = Yellow river (Tongde) (Harkins et al., 2007; Wang et al., 2022), In = Indus River (near Nanga Parbat) (Ali and De Boer, 2010; Burbank et al., 1996; Zeitler et al., 2001). Data of elevation and crustal thickness are from ETOPO1 (NOAA, 2009) and Crust 1.0 model (Laske et al., 2013), respectively.

Our models demonstrate that valley incision into continental crust can result in recognizable crustal deformation. In scenarios with the highest erosion potential, this deformation manifests as plastic strain within the upper kilometers of the crust. Notably, our models reveal a deformation regime associated with this plastic deformation, primarily extensional in the central part of the valley (Figure 4b). This insight opens the possibility of comparing model-predicted deformation with that observed in natural incised valleys. However, such a comparison is not straightforward. Natural examples chosen in this study exhibiting the largest erosion potential

(Indus, Tsangpo, and Yellow rivers) are located in tectonically active regions. As a result, these valleys do not display extensional deformation at their centers, due to the overriding influence of convergent tectonic settings. Conversely, rivers like the Ebro and Potomac have relatively low erosion potential, likely insufficient to induce observable incision-related surface deformation as observed in our models (Figure S3). Finally, while the Grand Canyon does feature extensional faults in its center (Billingsley et al., 2019), these faults predate the onset of Colorado River incision and therefore cannot be attributed to the valley formation. Consequently, the shallow deformation observed in these natural settings cannot be unequivocally interpreted as a consequence of the incision of their valleys.

#### **5 Conclusions**

We performed a series of thermo-mechanical numerical models including surface processes to investigate the deformation and exhumation of a thick continental crust subjected to valley incising an elevated plateau. Our results show that the incision of a single localized, but wide, valley is sufficient to generate significant crustal deformation, ultimately leading, for some specific conditions, to the exhumation of lower crustal material. This exhumation is enabled by the development of an upward crustal flow beneath the valley center, which facilitates the rise of low-viscosity material towards the surface and promotes lower crustal exhumation.

Our results indicate that lower crustal exhumation is only possible within a 10 Myr timeframe if favorable conditions are met, such as high elevation plateau (≥ 2 km), thick crust (> 50 km), and a long-term erosion rate (> 0.5 mm.yr<sup>-1</sup>). Achieving exhumation within 5 Myr requires more restrictive conditions, with elevation plateau

over 3 km, crustal thickness over 55 km and a long-term effective erosion over 0.8 mm.yr<sup>-1</sup>.

We also show that the parameters controlling the onset of lower crustal exhumation differ from those governing the final quantity of exhumed material. While the onset of exhumation is primarily driven by the initial plateau elevation, the final amount of exhumed lower crust is largely controlled by the initial crustal thickness. An increase in crustal thickness leads to a thicker lower crust, which enhances the upward flow of material and ultimately increases the volume of exhumed lower crust.

Last, we show that our models can be informative when compared to natural examples, even though our model set-up does not perfectly replicate natural examples, particularly regarding regional tectonic boundary conditions.

These first order results demonstrate that our model outcomes are broadly consistent with observations from natural systems.

ž

**Appendices** 

### Appendix A: Rheological model

This appendix detailed the equation of each component of the strain rate solved in the 2D thermo-mechanical code

The viscous strain rate  $\dot{\varepsilon}_{ij}^{\ v} = \dot{\varepsilon}_{ij}^{\ dis} + \dot{\varepsilon}^{\ dif} + \dot{\varepsilon}_{ij}^{\ Peierls}$  accounts for the dislocation, the diffusion and the Peierls creep mechanism, represented by dis, dif and Peierls exponents, respectively. The dislocation and diffusion creep strain rate ( $\dot{\varepsilon}_{ij}^{\ dis,dif}$ ) can be expressed as:

$$\dot{\varepsilon}_{ij}^{\text{dis,dif}} = \dot{\varepsilon}_{\text{II}}^{\text{dis,dif}} \frac{\tau_{ij}}{\tau_{\text{II}}} = A \left( 2f_{\text{dis,dif}} \exp\left(\frac{Q}{nRT}\right) \exp\left(\frac{-aM}{n}\right) \right)^{-n} d^{-m} \tau_{\text{II}}^{n} \frac{\tau_{ij}}{\tau_{\text{II}}}, \tag{A1}$$

where n, A, and Q are the dislocation creep parameters of the material (see Table 1), a is a melt weakening factor set to 0 in the lithospheric and astenospheric mantle and 50 in the crust for the partial melting models, M the melt fraction and R is the universal gas constant ( $R = 8.314510 \text{ J.K}^{-1}.\text{mol}^{-1}$ ). The subscript II indicates that this is the second invariant of the considered tensor.  $f_{\text{dis,dif}}$  corresponds to the correction factor (e.g., Schmalholz & Fletcher,

2011) for invariant formulation relative to the type of experiments used for calibration (here uniaxial compression),

such as:

$$f_{\text{dis,dif}} = \frac{1}{6} 2^{\frac{1}{n}} 3^{\frac{n-1}{2n}},\tag{A2}$$

Peierls strain rate is written:

603

$$\dot{\varepsilon}_{ij}^{\text{Peierls}} = \dot{\varepsilon}_{\text{II}}^{\text{Peierls}} \frac{\tau_{ij}}{\tau_{\text{II}}},\tag{A3}$$

with

$$\begin{cases}
\dot{\varepsilon}_{\text{II}}^{\text{Peierls}} = \left(2A^{\text{Peierls}}\right)^{-s} \frac{\tau_{ij}}{\tau_{\text{II}}} \\
A^{\text{Peierls}} = f\gamma\sigma^{\text{Peierls}} \left(E^{\text{Peierls}}e^{-\frac{(1-\gamma)^2Q^{\text{Peierls}}}{RT}}\right)^{\frac{-1}{s}}
\end{cases} (A4)$$

$$s = \frac{Q^{\text{Peierls}}}{RT} (1-\gamma)^{(q-1)q\gamma}$$

s is the effective stress exponent (*T*-dependent),  $Q^{\text{Peierls}}$  is the activation energy,  $\sigma^{\text{Peierls}}$  is the Peierls stress,  $E^{\text{Peierls}}$ , q (=2.0), and  $\gamma$  (=0.1) (Evans & Goetze, 1979) (Table 1). Peierls creep stress is computed using a regularised formulation (Kameyama et al., 1999)

Elastic strain rate is expressed following Hooke's law as:

$$\dot{\varepsilon}_{ij}^{e} = \frac{\dot{\tau}_{ij}}{2G} \frac{\tau_{ij}}{\tau_{IJ}},\tag{A5}$$

where G is the shear modulus (see Table 1) and  $\dot{\tau}_{ij}$  is the Jaumann derivative of the deviatoric stress.

The 2D simulations include the effect of frictional rheology. To this end a viscoplastic Drucker-Prager rheological model is employed. The yield function is expressed as:

$$F = \tau_{II} - P \sin \varphi - C \cos \varphi - 2\eta_{vp} \dot{\varepsilon}_{II}^{vp}, \tag{A6}$$

- where  $\tau_{II}$  is the deviatoric second stress invariant, P the pressure, C the cohesion and  $\varphi$  the
- friction angle. The viscoplastic parameter  $\eta_{vp}$  is set such that the overstress  $(2\eta_{vp}\dot{\epsilon}_{II}^{vp})$  is on the order of 1 MPa (i.e.
- $\eta_{\rm vp} = 10^{21} \, \text{Pa.s.}$ ) in our simulations.

#### Appendix B: partial melting implementation

This appendix detailed the partial melting equation solved in the 2D thermo-mechanical code MDoodz. In partial melting models, latent heat  $(Q_L)$  can be significant during partial melting or crystallization events. During melting,  $Q_L$  acts as a heat sink  $(Q_L < 0)$  whereas crystallization results in heat production  $(Q_L > 0)$ . In models including partial melting, this process is incorporated through: (i) a decrease in density with increasing melt fraction, (ii) the modification of effective viscosity and (iii) the consideration of the thermal impacts associated with melting/crystallization (Stüwe, 1995). The volumetric fraction of melt M is assumed to increase linearly with temperature, according to these relationships (Burg and Gerya, 2005; Gerya and Yuen, 2003):

 $M = 0, \text{ at } T \le T_{\text{solidus}}$ 646  $M = \frac{T - T_{\text{solidus}}}{T_{\text{liquidus}} - T_{\text{solidus}}}, \text{ at } T_{\text{solidus}} < T < T_{\text{liquidus}}$ 647  $M = 1, \text{ at } T \ge T_{\text{liquidus}}$ (B1)

 $M = 1, \text{ at } I \ge I_{\text{liquidu}}$ 

where  $T_{\text{solidus}}$  and  $T_{\text{liquidus}}$  are the wet solidus and dry liquidus temperatures of the rocks under consideration, respectively. The volumetric fraction of melt affects the effective density ( $\rho^{\text{eff}}$ ) of the partially molten rocks as follows:

$$\rho^{\text{eff}} = \rho \left( 1 - M + M \frac{\rho_0^{\text{molten}}}{\rho_0^{\text{solid}}} \right), \tag{B2}$$

where  $\rho_0^{\rm solid} = \rho_0$  and  $\rho_0^{\rm molten} = \rho_0 - \Delta \rho$  (see Table 1).

Other processes associated with partial melting, such as solid-melt segregation and melt extraction, are neglected.

The effect of latent heating due to equilibrium melting or crystallization is included implicitly by increasing the effective heat capacity ( $C_P^{\text{eff}}$ ) and the thermal expansion ( $\alpha^{\text{eff}}$ ) of the partially molten rocks (0 < M < 1)(Burg and Gerya, 2005).  $Q_L$  is the latent heat of melting of the considered rocks.

$$C_{\rm P}^{\rm eff} = C_P + Q_L \left(\frac{\partial M}{\partial T}\right)_P \tag{B3}$$

$$\alpha^{\text{eff}} = \alpha - \rho \frac{Q_L}{T} \left( \frac{\partial M}{\partial P} \right)_T$$
 (B4)

| Crustal partial melting parameters and equations | Crust                                                   |
|--------------------------------------------------|---------------------------------------------------------|
| $T_{\text{solidus}}$ (K) if $P < 1600$ MPa       | 973 - 70400/(P + 354) + 77800000/(P + 354) <sup>2</sup> |
| $T_{\text{solidus}}$ (K) if $P > 1600$ MPa       | $935 + 0.0035 \times P + 0.0000062 \times P^2$          |
| T <sub>liquidus</sub> (K)                        | 1423 + 0.105×P                                          |
| $\Delta \rho$ (kg.m <sup>-3</sup> )              | 200                                                     |
| а                                                | 50                                                      |
| H <sub>L</sub> [J.kg <sup>-1</sup> ]             | 380,000                                                 |

Table B1: Equations and parameters used to implement crustal partial melting. Equations for the liquidus and solidus temperatures are from Schmidt and Poli (1998) and Poli and Schmidt (2002), where P corresponds to the pressure in MPa. The value for the latent heat of melting  $(H_{\rm L})$  is in line with standard values used for crustal material (e.g.

Gerya, 2019; Turcotte & Schubert, 2002). The value of the change in density due to partial melting ( $\Delta \rho$ ) is in line with standard values, assuming a drop of density of 5-10% for partially melted granite (e.g., Philpotts & Ague, 2009). The a value used in Eq. A1 is set to produce a drop of viscosity of three orders of magnitude for 10% of partial melting.

### Code availability

The data of this study have been generated by using the code MDoodz7.0. This code is freely accessible and all the information needed to install the code and use it is available through Zenodo Geffroy et al., (2025a), including the updated files of the code specifically written for this study. The Matlab file used to read the Output files and produce the figures of this paper is also given in Geffroy et al., (2025a)

#### Data availability

All output data of the reference are through Zenodo (Geffroy et al., 2025b)

## **Authors contribution**

TD and PY developed the model code and TG performed the simulations. TG prepared the manuscript with contribution of all co-authors.

# **Competing interest**

Acknowledgements

The authors declare that they have no conflict of interest

This research has been supported by the Université de Rennes, CNRS and the H2020 European Research Council (grant agreement no. 803721) and the Institut Universitaire de France (IUF). P.Y. would particularly like to thank

 Jean Van Den Driessche for the discussions that initiated this work several years ago. Clément Cholet is also thanked here for having produced the first models of this type using MDoodz during his Master 2 internship. We

- also want to sincerely acknowledge the two reviewers, Carole Petit and Guillaume Duclaux for their insightful
- comments that provided valuable guidance in improving our study.
- Financial support

- This work was supported by the Institut Universitaire de France (IUF)

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
