# Peer review of "Deformation and exhumation in thick continental crusts"

_EGUsphere, 2025_

## Author Comment (AC1)

*'This is an original study on the effect of large valley incision on lower crust exhumation in the context of a continental plateau underlain by a thick continental crust with very low ductile resistance. The authors conclude that, under certain conditions (high plateau elevation, thick crust, large incision rate and prolonged incision history), river incision can lead to exhumation of the ductile lower crust beneath the valley axis. The article is well-written, has high-quality illustrations, and I agree with the main interpretation of the model results. However, I have several criticisms that should be addressed prior to publication, this is why I ask for major revisions although I think that they will be relatively easy to address. My comments to the authors are as follows'*

Dear Reviewer#1,

We sincerely appreciate your valuable feedback on our manuscript entitled "Deformation and exhumation in thick continental crust induced by valley incision of elevated plateaux." Your insightful comments have provided valuable guidance in improving our study and sharpening the clarity of our arguments. Below, we provide a detailed answer for each of your comments. Modifications added in the manuscript are also shown:

- *'I understand that in nature, crustal thickness and plateau elevation can vary widely. However, in your case, since you assume constant crust and mantle densities, there should be a linear relationship between these two parameters due to isostasy. This means that you cannot arbitrarily choose both crustal thickness and plateau elevation independently. For example, if we assume a mean crustal thickness of 35 km for sea-level elevation, then local isostasy (neglecting density changes due to temperature and pressure and including the plateau in the total crustal thickness) would give approximately the following values: a 1 km-high plateau corresponds to a total crustal thickness of 42 km, 2 km corresponds to 49 km, and 3 km to 56 km. How, then, can you justify a 65 km-thick crust with a 3 km-high plateau, as shown in Figure 2a, using your chosen densities, without introducing a significant initial isostatic imbalance? Am I missing something here?'*

We agree with the reviewer that, assuming a reference elevation of 0 km for a 35 km thick continental crust and using the densities implemented in our models, there is a linear relationship between crustal thickness and plateau elevation. This relationship would typically prevent testing different plateau elevations for a given crustal thickness. However, in our models, the surface of the continental crust is initially flat and does not experience lateral pressure gradients that would drive isostatic re-equilibration. Although we tested various plateau elevations for the same crustal thickness, our models (black dots in Figure 2a) align closely, in particular the reference model, with the expected linear relationship between crustal thickness and plateau elevation whatever the chosen crustal thickness. This approach allowed us to more comprehensively explore the range of plateau elevation/crustal thickness combinations observed in nature. Relying strictly on the isostatic relationship would have limited our ability to disentangle the individual effects of crustal thickness and plateau elevation on model behavior as the isostatic linear relationship would lead to only one possible elevation for each crustal thickness.

To clarify and justify this modeling choice, we now include the theoretical isostatic relationship in Figure 2a and have added a corresponding explanation in the Methods section 2.3 (line 219 to 228).

- *'I find one of your results particularly interesting — that surface uplift can be decoupled from the lithospheric response, and that crust-mantle decoupling explains why the Moho remains stable while the lower crust migrates toward zones of lower pressure. Could this behavior explain the very low effective elastic thickness often inferred from the isostatic response of the lithosphere to surface processes? In other words, could this be explained by a situation where only the upper crust effectively responds?'*

We agree with the reviewer's comment that the decoupling between the crust and the mantle in our models lead to a scenario in which only the crust responds to valley incision. This decoupling would affect the calculation of the effective elastic thickness (EET), as the response would reflect crustal behavior alone rather than that of the entire lithosphere. Consequently, the computed EET in the context of erosional processes in orogenic settings would be lower, potentially explaining the systematically low EET values observed in such regions. To fully address this point, we have added a dedicated discussion section 4.2 in the manuscript (line 453 to 463):

- *'Yield stress envelope in Figure 2b: there appears to be no strength in the mantle, which seems surprising. With the dry olivine rheology you use, I would expect some resistance.'*

The absence of strength in the mantle is consistent with the thermal gradient chosen in our models. At the crust-mantle boundary, temperature is ~ 1100 °C. The figure below shows the computed differential stress for the dry olivine rheology (from Hirth & Kohlstedt, 2003) as a function of temperature. In such a configuration, it is normal to have a very low strength in the mantle (see Figure 1 below).

[Figure]

**Figure 1: Differential stress evolution as a function of temperature for the Dry Olivine rheology.**

- *'I'm generally not in favor of requesting additional model runs in modeling papers, as this can easily become an endless process. However, I am somewhat puzzled by the fact that you don't really discuss your choice of an extremely weak and thick lower crust, which leads to strong convection and very rapid ductile flow. While I understand this may be a deliberate choice, I think it would be helpful to include a comment on how this specific rheology — which possibly resembles that of an orogenic crust — may not represent the "average" continental crust. Out of curiosity, I would be very interested to see how the system behaves with a more resistant (mafic) lower crust and/or a colder lithosphere. For instance, you could add another dimension to your parameter space in Figure 11, for instance by representing the*

*effect of the thickness and/or average viscosity of the ductile crust and the comparison to natural settings.'*

**In order to address this comment, we performed two additional models: one using the same rheology but with a colder thermal structure (temperature at the Moho of 800 °C), and another employing a diabase rheology for the lower crust. Both models have the same plateau elevation and crustal thickness as the reference case. The methodologies and figures for both models are presented in the supplementary materials (Figs. S8 and S9), and their results are now discussed in section 4.2 (line 481 to 489).**

- *'By the way, you should clearly define in the main text what you mean by "lower crust." In your model, you designate crust below 10 km depth as the lower crust, but it shares the same rheological properties as the upper crust. In the literature, "lower crust" can refer either to the ductile portion of the crust — as you do here — or to the more mafic and mechanically stronger part of the continental crust. While this is briefly explained in a figure, it would be helpful to clarify this choice explicitly in the main text to avoid confusion.'*

**You are right. We now give additional information on the definition of the lower crust in our models in the Method section 2.2 (line 184 to 186).**

- *'Along the same lines, I'm not sure that such an overthickened and weak crust could remain stable without collapsing, unless it's being artificially supported by the model boundaries. This issue is not visible in your setup because you impose a constant crustal thickness and therefore remove any lateral pressure gradients (except the ones due to valley incision). But from a large-scale geodynamic perspective, the configuration might not be entirely realistic — especially if we consider that real-world plateaus are not laterally infinite. That said, since your model already shows lower crustal flow driven solely by valley incision, I can only imagine how much flow would occur if this plateau were adjacent to a region of much lower elevation and much thinner crust.'*

**We agree with the reviewer's comment that, in a natural setting, if a plateau is adjacent to a region of significantly lower elevation, the flow of lower crustal material would preferentially migrate toward this area, potentially altering the overall lower crustal response to valley incision. However, in our modeling strategy, we deliberately choose to isolate the effect of valley incision itself, independently of any surrounding topographic or tectonic gradients. To achieve this, we imposed fixed lateral boundaries, preventing material from flowing out of the model domain.**
**Furthermore, our simulations indicate that the lower crustal material mobilized by valley incision originates from a relatively narrow area around the valley. To illustrate this, we performed a model in which the total width of the model domain was reduced from 500 km (reference model) to 200 km (see Figure 2 below).**

[Figure]

*Figure 2: Material and thermal distribution after 10 Myr for a 500-km wide model (left) and a 200-km wide model (right).*

The results show that the overall crustal response, as well as the timing and magnitude of lower crustal exhumation, remain very similar to those observed in the original 500 km-wide reference model.

---

## Author Comment (AC2)

*Review of "Deformation and exhumation in thick continental crusts induced by valley incision of elevated plateaux", by Thomas Geffroy, Philippe Yamato, Philippe Steer, Benjamin Guillaume, and Thibault Duretz.*

*This paper presents a comprehensive numerical study of the impact of valley incision on crustal deformation. Using coupled 2D thermo-mechanical & surface process models the authors present the evolution of a hot and weak crust topped with an orogenic plateau subjected to constant river vertical incision down to a predefined base level. A total of 48 models have been used to explore the role crustal thickness, initial plateau elevation and incision velocity in controlling the relief evolution, crustal exhumation and strain distribution in the crust. Although the authors insist the model aims to reflect tectonically inactive regions, the settings tested (especially where crustal thickness is ≥ 50km, that is for all but 9 models) appear more representative of orogenic plateaux or orogenic systems in general. The comparison with natural examples in the Nanga Parbat and Namche Barwa Massifs in the introduction points that way.*

*The paper briefly reviews published literature on the role of valley incision, insisting on the role of erosion potential in controlling crustal deformation over long periods of time (here up to 10 Myr). I really liked the synthesis proposed in Fig 2a showing some statistics about crustal thickness and surface elevation. The physical description of the model is detailed in the paper and the appendices, but needs to be updated here and there (see details below). Results and discussion are well organized but the importance of the partially molten lower crust (and the very high geotherm) is not discussed in enough details, in particular with respect to the exhumation pattern.*

*Overall this contribution is of broad interest and has the potential to create impact in both the tectonic and geomorphology communities, worth publishing in Solid Earth journal. The manuscript is well written and nicely illustrated, and some reworking should make this a solid contribution. The references seem adequate too. I would recommend accepting this manuscript after moderate revisions.*

*Below, I outline specific points for improvement, ranging from minor corrections to more critical issues:"*

**We sincerely appreciate the valuable feedback from reviewer#2 on our manuscript. His insightful comments have provided valuable guidance in improving our study and sharpening the clarity of our arguments. Below, we provide a detailed answer for each of the comments. Modifications added in the manuscript are also shown (*in italic*):**

*+ line 35: you mention that "first order results align well with observations from natural systems", please provide at least a couple examples here in the abstract. The comparison with natural systems is one of the main concern I had when reading the manuscript, so this should be strengthen. Based on what you provide in the introduction (l. 74-75) it is not clear to me the setting really applies for "inactive regions".*

**You are correct. We have now included natural examples that we consider to be in good agreement with our model results in the Abstract (line 34 to 37).**

Regarding the second part of the comment, it is important to clarify that our model settings are not designed to represent "tectonically inactive regions", but rather to isolate the crustal response to valley incision. To achieve this, we designed a setup that excludes horizontal velocities at the lateral boundaries of the models. We acknowledge that the use of a significant crustal thickness implies a tectonically active context, but including shortening as boundary conditions in the model would have prevented us from isolating the effects of valley incision on crustal deformation. In order to clarify this point, we now added few lines in the Introduction section (line 80 to 85).

*+ Line 73-74: Capitalize "Massif" (e.g., "Nanga Parbat Massif")*
**Done**

*+ line 80: you briefly mention the type of constitutive laws used in the thermo-mechanical model, but there is no detail regarding the surface processes here. I suggest adding that the surface process model is a simple erosion law coupled with diffusion. More details are provided later l137 to 144.*

**Done: Line 87 to 88:**

*+ line 83: Why did you choose this 10 Myr cutoff value? I would assume that in inactive regions processes can be much slower... supposedly the river incision velocity wouldn't be as large as tested in your experiments.*

We selected a value of 10 Myr as an upper bound for the duration a river may flow along the same path. As illustrated in Figure 11b, river persistence at a fixed location over a timescale of 5 Myr or more are not very numerous. Moreover, natural examples of valley incision associated with lower crust exhumation indicate that such exhumation typically occurs within less than 10 Myr. Therefore, we assume that if this process does not take place within a 10 Myr timeframe in our model, it is unlikely to be geologically realistic. A justification has been added to the manuscript in the Introduction section to clarify this assumption (Line 93 to 96).

*Regarding the equations in general, please use bold symbols for vectors and tensors.*

Thank you for pointing that out. The initial version of the manuscript did include some inconsistencies and lacked clear definitions for certain terms in the equations. However, we decided to keep our nomenclature of vector and tensor, and added the different component of these vectors and tensor in the manuscript

*+ line 99, eq 1: g_i --> should be g, the gravity acceleration as defined l. 104*

The two components of $g_i$ are now described (Section 2.1).

*+ line 100, eq 2: v_i, this term is not defined. Please mention v_i or rather *v* is the velocity.*

**Done : Section 2.1: "$v_i$ the velocity component in the direction i"**

*+ line 101, eq 3: C_P^{eff} is written with an upper case P (as for the equations in Appendix B), but it is spell with a lower case p line 104. Fix it on line 104. There is another major issue with the Heat source terms Q_L and Q_r in Equation 3 which should be in W.m^-3 but they are provided in J.kg^-1 (Table B1, l606) and W.m^-2 (Table 1, p 7) respectively. Please fix the units and make sure you've used appropriate values.*

**The letter P not in upper case has been changed. The unit of the $Q_r$ has been changed in the Table 1 to W.m$^{-3}$. $Q_L$ is not heat source term as it corresponds to latent heat energy. It has been removed from the equation 3, as it is already used for the calculation of $C_P^{eff}$ (see Eq. B3). To facilitate the understanding, we renamed $Q_L$ in $H_L$. All the terms are now consistent.**

*+ Figure 2: I am totally frustrated not to see the geotherm plotted along the vertical strength profile. Moho temperature in all these models is super high (>1000°C) and important for explaining the convective regime in the partially molten lower crustal domain. I suggest you provide this information for the different models. In fact in Figure 4 it seems that T_Moho is decreasing through time.*

**We followed the reviewer's recommendation and added the temperature profile to Figure 2. Additionally, we provide the temperature profiles corresponding to each snapshot of the reference model shown in Figure 4, now included in the supplementary material (Figure S2).**

*+ Table 1: You provide here diffusion creep parameters for the mantle, yet in Appendix A you write that viscous strain is the sum of dislocation and Peierls crop only. There is not mention of diffusion. Please revise the Appendix and you should also mention that Peierls creep is apply to the lithospheric and asthenospheric mantle materials only.*

**You're right. We added the diffusion creep component in equations of the Appendix, and mention that these laws apply to both the lithosphere and the asthenosphere.**

*+ line 209: you mention the computation of the effective erosion rate (E_eff), could you precise what discretization is applied through time for this calculation?*

**The discretization of the calculation of $E_{eff}$ is not uniform over time, as $E_{eff}$ is computed every 2 km of crustal material crossing the valley center. This specification has been added in Section 2.4 (Line 240 to 241).**

*+ line 233: the time thresholds provided appear to be exact round numbers... Is that because of the model time stepping or output intervals?*

**The given values are actually rounded values presented in the manuscript with the sign "~". Reporting more digits after the decimal point offers no significant benefit. We therefore preferred to present rounded values for more fluidity in the manuscript.**

*+ line 253-257: To me the "two distinct high strain rate zones" are not clearly visible. Strain appears diffuse in the viscous crust (between ~6km and the partially molten region). I suggest you provide close up views of these objects, zooming in the crustal region where $T < \sim 650\,°C$. As it appears now the lower crustal convection attracts most of the attention.*

**We chose to display the first 100 km of the model to capture the strain rate evolution occurring not only in the crust but also in the underlying mantle. Focusing solely on the non–partially melted portion of the crust would prevent the reader from observing the processes taking place in the mantle. However, to improve clarity, we have reformulated the corresponding sentence as follows in the Results section 3.1.2 (Line 285 to 290):**

*+ Figure 4: It would be nice to have the melt fraction in the crust displayed, either with contours or along a vertical profile as you did for the for the strength profile in Fig. 2. The lower crust temperature rises more than 300K over the solidus so I suspect there is a quite large melt fraction... is that reasonable to assume?*

**You are right. We now present the results of the melt fraction. The melt fraction profiles at the center of the valley for all model snapshots presented in Figure 4 are now provided in the supplementary material, alongside the corresponding temperature profiles (see Figure S2). The following sentence was modified accordingly in Section 3.1.2 (Line 299 to 301):**

*+ lines 277-278: I'm a little confused here. If the plateau height can vary independently from the crustal thickness it means the model isn't at isostatic equilibrium. That seems like a problem for an inactive tectonic setting. Because of the left and right BC applied to the models I assume any configuration will be "stable", the free-slip BCs kind of mimic a lateral stress. Another way to write this is that the assumption of isostatic equilibrium is unclear. The free-slip boundary conditions may artificially stabilize the system, warranting further discussion. As such I have some doubts about the reasoning in this section.*

**The first part of the reviewer's comment has already been addressed in our response to the Reviewer#1. We agree that, under isostatic considerations, plateau elevation and crustal thickness should be directly linked by a linear relationship. This dependency would generally preclude testing different plateau elevations for a given crustal thickness. In response to this point, we now include the theoretical isostatic relationship in Figure 2a and provide a corresponding explanation in Methods section 2.3 (Line 219 to 228).**

**However, we respectfully disagree with the reviewer's comment regarding the artificial stabilization of the system. In our models, the initial topography is flat, which indeed prevents any initial re-equilibration. However, if the initial topography was not flat, the system would naturally re-equilibrate, because even if there is no horizontal velocity applied, both left and right sides are free-slip boundary conditions. This behavior can be observed in our simulations, where the topography adjusts following the onset of incision and the subsequent development of relief.**

*+ line 295: why is the threshold limit exactly 3km for h_P? Is this number related to the model configuration (i.e. model-dependent)? Could you please elaborate on this in the discussion?*

Since in models with $h_P < 2$ km the valley reaches the base level, and in models with $h_P > 3$ km it does not, we define the threshold value at 3 km. Although the actual threshold likely lies between 2 and 3 km, we do not simulate intermediate values; therefore, we adopt 3 km as the reference threshold distinguishing whether or not the valley reaches the base level. To clarify this point, we have included a new figure in the supplementary material (Figure S3) that shows the time at which base level is reached, or not, for each model (called in Section 4.1).

*+ line 347: "after 1-2 Myr" --> it seems $\Delta h$ reaches a plateau as soon as h_min hits the base level before 1 Myr. After that it gently increases and slightly oscillates. Could you rephrase this part?*

We agree with the reviewer's comment and have rephrased this part  (Line 382 to 384)

*+ line 374: Could you please explicitly cite in the manuscript which of the models have the valley reaching the imposed base level?*

Yes, following your advice and in line with your previous comment, we now refer to Figure S3, which highlights the time at which each model reaches base level.

*+ section 4.2. In the model presented here the upper crust if completely decoupled from the mantle because of the partially molten lower crust. Moreover the Moho appears flat for every model. Line 420 you propose that in the models presented vertical flow is associated with crustal isostatic rebound rather than lithosphere response. I would argue that the bottom BC applied in the model doesn't allow for spatial variable lithosphere readjustments. So the discussed behavior appears to be a feature of the model and I believe you simply can't compare your results with Vernant et al. (2013).*

We respectfully disagree with the reviewer's comment suggesting that the bottom boundary condition would prevent lithospheric readjustment in response to valley incision. At this temperature (i.e., 1330 °C), the strength of dry olivine is negligible, allowing the lithosphere to deform freely if required (see Figure 1 below).

[Figure]

**Figure 1: Differential stress as a function of temperature for the Dry Olivine rheology**

Moreover, we consider that sufficient depth is provided in the model to allow both the lithosphere and asthenosphere to deform freely. To support this point, we conducted an additional simulation with a domain thickness of 300 km, offering significantly more space for lithospheric adjustment. This model shows no significant differences compared to the reference model with a 150 km depth, indicating that the bottom boundary condition does not impede lithospheric readjustment (see Figure 2 below).

[Figure]

**Figure 2: Material and thermal distributions after 10 Myr for a 150-km thick model (left) and a 300-km thick model (right).**

*+ line 483-487: I admit am not very familiar with the geological history of the Grand Canyon, but from what I recall there is a long protracted evolution of the Early Proterozoic basement predating the emplacement of the Colorado plateau (e.g the Vishnu schists). It seems evident that the lower crust material eroded during the plateau incision were already near the surface before Canyon incision took place.*

**We agree with the reviewer's comment, and added this specific point in the discussion regarding the Grand Canyon (Line 538 to 541)**

*+ line 517: "performed a series of thermo-mechanical AND SURFACE PROCESSES numerical models"*

**We agree. Done**

Appendices

*+ line 542: add \dot\varepsilon_{ij}^{diff} term for the viscous strain as you have this process for mantle rocks, along with Peierls.*

**Done**

*+ line 549-550: replace "second tensor invariant" with "second invariant of the strain rate tensor".*

**We rephrase this sentence as follows: "*The subscript II indicates that this is the second invariant of the considered tensor*"**

*+ line 551: I believe you meant uniaxial rather than axial.*

**Done**

*+ line 567: misspelled "deviator stress. Remove final "e".*

**Done**

+ Table B1: replace Mpa with MPa in the first column.

**Done**